# Half of resources in threatened species conservation plans are allocated to research and monitoring

Rachel T. Buxton[1]✉, Stephanie Avery-Gomm [2], Hsein-Yung Lin[1], Paul A. Smith[1,2], Steven J. Cooke[1,3] & Joseph R. Bennett [1,3]

Funds to combat biodiversity loss are insufficient, requiring conservation managers to make trade-offs between costs for actions to avoid further loss and costs for research and monitoring to guide effective actions. Using species' management plans for 2328 listed species from three countries we show that 50% of species' proposed recovery plan budgets are allocated to research and monitoring. The proportion of budgets allocated to research and monitoring vary among jurisdictions and taxa, but overall, species with higher proportions of budgets allocated to research and monitoring have poorer recovery outcomes. The proportion allocated to research and monitoring is lower for more recent recovery plans, but for some species, plans have allocated the majority of funds to information gathering for decades. We provide recommendations for careful examination of the value of collecting new information in recovery planning to ensure that conservation programs emphasize action or research and monitoring that directly informs action.

[1] Department of Biology, Carleton University, Ottawa, ON K1S 5B6, Canada. [2] Environment and Climate Change Canada, National Wildlife Research Centre, Ottawa, ON K1S 5B6, Canada. [3] Institute of Environmental and Interdisciplinary Science, Carleton University, Ottawa, ON K1S 5B6, Canada. ✉email: Rachel.buxton@colostate.edu

Given rapid rates of biodiversity loss and limited funding, recovery programs face difficult decisions about which conservation actions are the highest priority. Management of threatened species requires trade-offs between action and information: conservation actions are necessary to secure species from extinction, but management decisions made in the absence of sufficient information can be inefficient, or worse, undermine progress towards recovery[1]. Decisions must be made for threatened species with remarkably insufficient resources—in the U.S. for example, expenditure on endangered species is only 25% of that needed for full implementation of recovery plans[2,3]. Thus, conservation budgets represent a challenging resource allocation problem, where managers must efficiently balance the costs and benefits of management actions[4] with the value of collecting further information to increase the certainty of management success[5,6].

Research and monitoring (RM) are important components of threatened species conservation. We define RM as activities that generate information about species (e.g., ecology, trends, population biology), threats they face, the socioeconomic context in which they occur (e.g., competing land uses), their response to interventions, and the effectiveness of new management techniques, including information designed to improve management approaches[7]. RM can lead to improved conservation decisions for threatened species when systematically integrated to iteratively improve the outcomes of management interventions (i.e., adaptive management) or may guide the implementation of actions based on the state of species populations i.e., state-dependent management[8]. In this way, when applied to inform action, RM can lead to improved efficiency and feasibility of management[9]. However, non-strategic or unwarranted RM can waste limited conservation resources, and reduces the funding available for action[10]. Prioritizing funding for RM may create the illusion that something useful is being done[11], allowing necessary but difficult decisions regarding management actions to be deferred. Worse, some conservation monitoring programs track populations without any plan for action if a change occurs or collect information with no immediate relevance to management decisions[12]. As a result, many local populations and species have been monitored until extinction[13,14].

Previous work has examined the cost of threatened species recovery plans as a resource allocation problem—optimizing the trade-off between the expected benefits and costs of management[15–17]. Yet, achieving recovery is unlikely if most resources are allocated to RM without clear guidance about how the information collected will trigger management interventions for recovery. We examine the proportion of threatened species budgets allocated to RM for over 2300 threatened species from three countries. We explore the characteristics of species with a high proportion of funding spent on RM to identify recovery plans that may not be sufficient to achieve recovery. We examine whether the proportion of the recovery budget allocated to RM is associated with threatened species recovery outcomes. Finally, we offer recommendations for examining the value of collecting new information when updating recovery documents, to ensure that research and monitoring are designed to generate evidence that can directly inform species recovery and increase the efficiency and effectiveness of future recovery strategies. Our findings show that, on average, half of species' proposed recovery plan budgets are allocated to RM and that species with higher proportions of the budgets allocated to RM have poorer recovery outcomes.

## Results

**Proportion of species' budgets allocated to RM.** Collectively, the United States (U.S.), New Zealand (NZ), and New South Wales, Australia (NSW), designate a mean of $50 \pm 27\%$ (±standard deviation, sd) of threatened species proposed budgets to RM. For 4% of species (3% in the U.S., 6% in NZ, and 2% in NSW), >95% of the proposed budget was allocated to RM (Fig. 1). For a subset of U.S. management tasks classified according to IUCN criteria, we found that the most common type of RM was an investigation of life history and ecology and the least common was research and monitoring of harvest and trade (Supplementary Fig. 1).

**Factors affecting the proportion allocated to RM.** The U.S. and NZ had a significantly higher proportion of species' budgets allocated to RM than NSW (mean ± sd, $52 \pm 24\%$ in the U.S., $52 \pm 28\%$ in NZ, and $36 \pm 28\%$ in NSW; Fig. 1, Supplementary Table 2). The proportion of the budget allocated to RM was lower for species where the predicted benefits of the actions contained within recovery plans were estimated to be higher (see Methods for details, Supplementary Table 2). This trend was less pronounced for threatened species plans in the U.S. (Supplementary Table 3), which may relate to how the relative benefit of implementing a recovery plan was estimated by Gerber et al.[15] (see Supplementary Methods). Across all jurisdictions, threatened species with a larger total proposed budget had a lower proportion of the budget allocated to RM (Supplementary Table 2). For species with a smaller total proposed budget, there was a large variation in the proportion of the budget allocated to RM (0–100%; Supplementary Fig. 3). Bryophytes had the highest proportion of the budget allocated to RM, but these species are only listed in NZ (Fig. 2, Supplementary Table 2, Supplementary Fig. 2). In the U.S. and NSW, amphibians had the highest proportion of the budget allocated to RM (Supplementary Fig. 2, Supplementary Table 2). Across all jurisdictions, birds had the lowest proportion of the budget allocated to RM (Fig. 2, Supplementary Table 2).

We explored additional characteristics unique to U.S. recovery documents, including species listing status, the proportion of RM management tasks noted as complete, the first fiscal year of the earliest RM, the number of species in the recovery plan, the proportion of RM assigned as high priority, and a covariate called recovery potential (see Methods, Table 1). Of these variables, only the proportion of RM assigned as high priority and the first fiscal year of the earliest RM were significant, whereby species with a higher proportion of RM assigned high priority and those where RM began longer ago had a higher proportion of the proposed budget allocated to RM (Supplementary Table 3). In addition, using U.S. species recovery plans, we summarized the proportion of budget allocated to RM by year the recovery plan was published and found that older plans had more resources allocated to RM than newer plans (Supplementary Fig. 4).

**Species recovery outcomes.** For species where an index of recovery could be extracted (79% of U.S. species, 14% of NZ species, and 15% of NSW species), those with the highest proportion of the budget allocated to RM had the lowest recovery success (Fig. 3). For example, in the U.S., species with a recovery index of −9 to −11 (indicating a declining status in 9 to 11 of 11 status reports) had a median of 70% of the proposed budget allocated to RM (Fig. 3). In NZ and NSW, species with a recovery index of −2 to −3 (indicating a declining status in 2 to 3 of 4 and 5 status reports, respectively) had a median of 44% of the proposed budget allocated to RM (Fig. 3).

## Discussion

On average, approximately half of all proposed budgets for threatened species recovery are allocated to research and monitoring. This percentage is significantly higher than research and

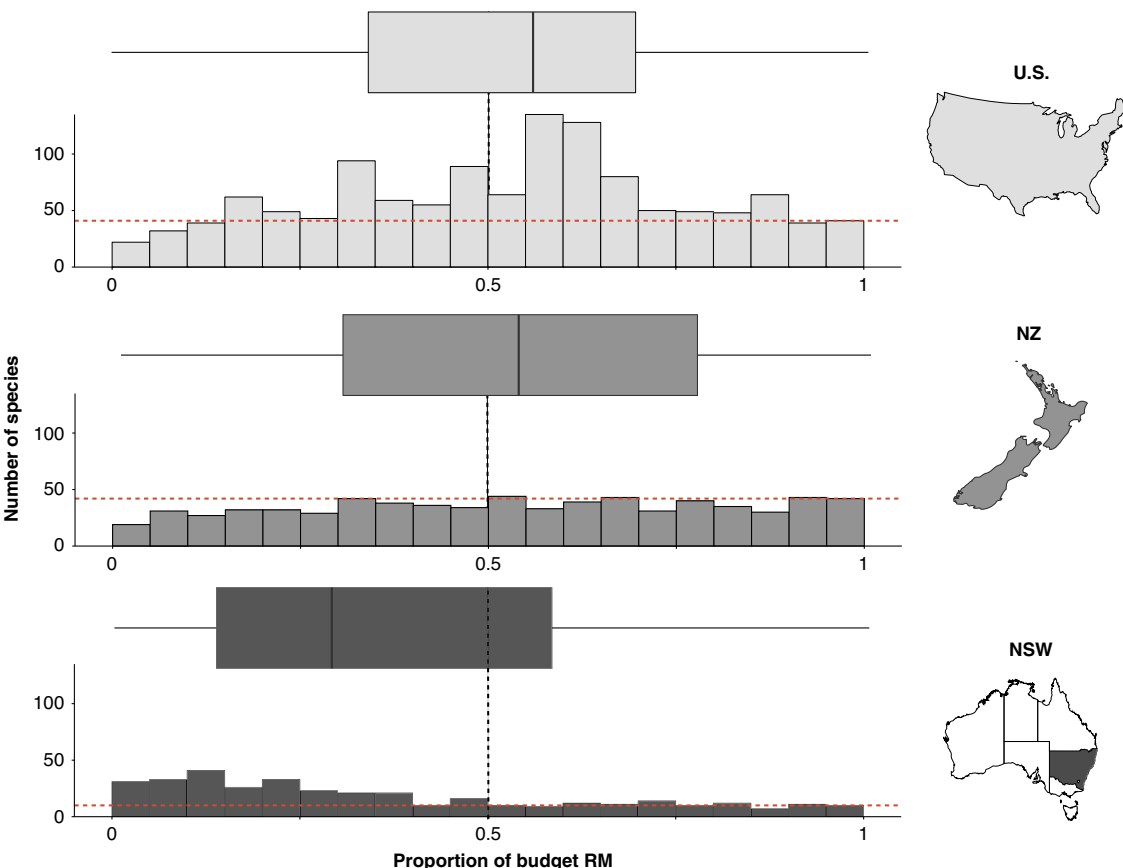

**Fig. 1 Proportion of the budget allocated to research and monitoring for threatened species ($n = 2261$ species in three jurisdictions: United States (U.S.), New Zealand (NZ), and New South Wales (NSW)).** The stippled red line indicates the number of species with >95% of the budget allocated to research and monitoring (RM). The box and whiskers show the proportion of recovery plan budgets allocated to research and monitoring in each jurisdiction, with the median as a line, first and third quartiles as hinges, and the highest and lowest values within 1.5 times the inter-quartile range as whiskers. The black stippled line indicates the mean among jurisdictions (50%). Maps were created in ArcGIS for Desktop (10.3, ESRI Inc., USA).

development (R&D) costs in other sectors: the top 10 largest corporations spend ~13% of annual revenue on R&D[18], and the pharmaceutical industry, which invests the most in R&D of any industry[19], spends on average 8–25% of its annual revenue on R&D initiatives[20]. We note that this comparison is not direct—conservation does not typically generate revenue—and percentages would be considerably different if RM were compared to contributions of threatened species to human society, which are consistently undervalued[21]. The difference between RM for threatened species and R&D in other sectors could be interpreted as indicative of high uncertainty in ecology[22]; however, complex decision-making with high stakes and large uncertainties are not unique to conservation biology (e.g., law[23], medicine[24], economics[25]). If planning to allocate half of conservation resources to RM is problematic, the reality may be more so. For most threatened species, only a small proportion of the total proposed budget is implemented[2], and only a fraction of proposed management tasks are achieved[26]. Thus, depending on the order in which tasks in the recovery plan are implemented, the proportion of resources allocated to RM could be much higher than described here.

Across all jurisdictions, we found that threatened species with poorer recovery outcomes had higher proportions of their recovery budgets allocated to RM. This relationship is likely a result of several factors. First, it suggests that planning almost exclusively for RM with little plan for action in recovery strategies is unlikely to abate threats and improve species status. Second, greater allocation of resources to RM for species with poor

recovery outcomes could suggest that high uncertainty associated with actions for especially imperiled species reinforces a fear of negative outcomes and may deter necessary actions[27]. Thus, there may be a predisposition to spend more on RM instead of action on species that are more critically endangered. Alternatively, species with worse recovery outcomes may require higher proportions of RM because little may be known about them and their threats. Regardless, the question remains: would allocating a greater proportion of funds to action improve recovery outcomes and if so, what is the optimal allocation between RM and action to maximize the achievement of conservation objectives? Other studies have shown that recovery outcomes are positively related to the number of years listed[28], years with a recovery plan[29], and funding[30], yet these effects are weak, potentially due to the low quality of species recovery data[28]. Gerber[2] found that spending is insufficient for the US Endangered Species Act (ESA), resources are allocated disproportionately among species, and there are significant discrepancies between proposed and actualized budgets, whereby excess budgets do not translate into better recovery outcomes. Thus, making deliberate decisions about resource allocation between species and potentially between RM and action offers the potential to improve outcomes for threatened species.

For some species, our results suggest that recovery programs may be trapped in a cycle where more resources are allocated to information gathering versus action. Among threatened species in the U.S., we found that when RM began longer ago there was a higher proportion of the budget allocated to RM, perhaps

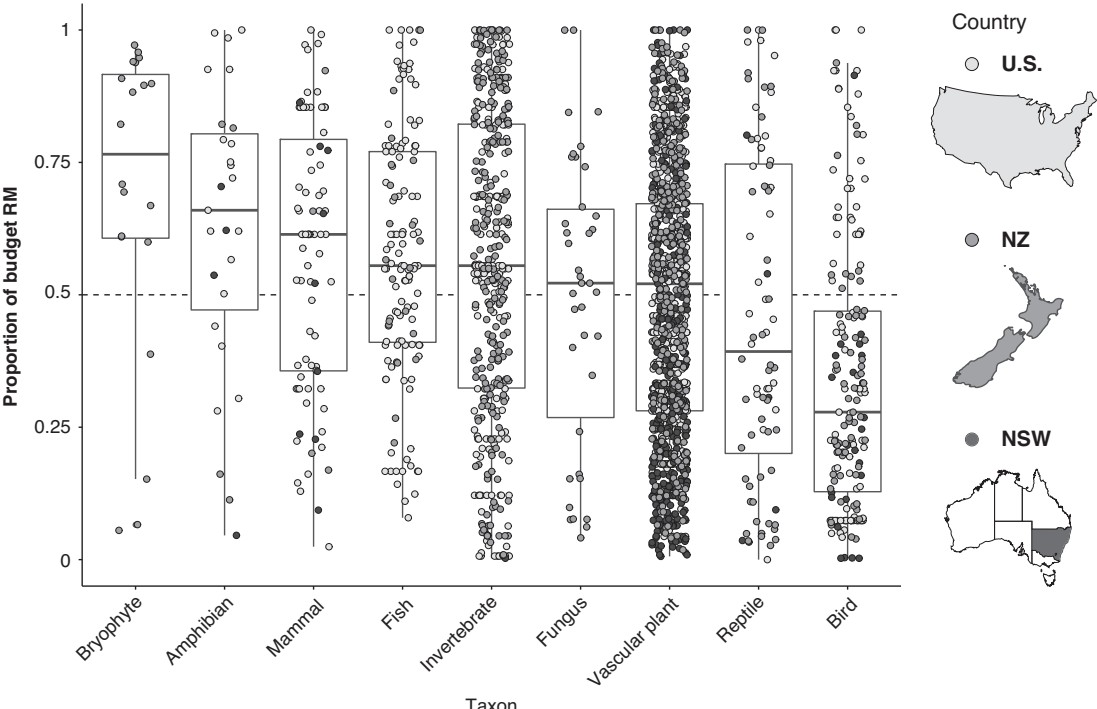

**Fig. 2 Proportion of the budget allocated to research and monitoring for threatened taxa.** The median and range of proportion of recovery plan budgets allocated to research and monitoring (RM) for $n = 2261$ threatened species in a variety of taxa in the United States (U.S.), New Zealand (NZ), and New South Wales, Australia (NSW). For each taxon, the box and whiskers show the median as a line, first and third quartiles as hinges, and the highest and lowest values within 1.5 times the inter-quartile range as whiskers. Maps were created in ArcGIS for Desktop (10.3, ESRI Inc., USA).

**Table 1 Covariates used to examine the drivers of proportion of threatened species budget allocated to research and monitoring (RM; in three jurisdictions: New Zealand, NZ, New South Wales, Australia, NSW, and the United States, U.S.).**

| Variable | Type | Details | Jurisdictions |
|---|---|---|---|
| Proportion of budget RM | Response | Proportion of cost of all management tasks for each species allocated to RM over a 50-year period | All |
| Benefit | Scaled covariate | Probability of species being secure in 50 years with all management tasks - without management tasks (NZ, NSW: expert elicitation, U.S.: generated from Recovery Priority Number) | All |
| Total budget | Scaled covariate | Previously published estimates of total cost of management tasks per species | All |
| Taxon | Dummy covariate | All: Amphibians, birds, bryophytes, fishes, fungus, invertebrates, mammals, reptiles, and vascular plants (reference category); U.S.: Amphibians, birds, fishes, invertebrates (reference category), mammals, reptiles, flowering plants, and non-flowering plants (lichens removed) | All |
| Status | Dummy covariate | Endangered, Threatened, or Not Listed | U.S. |
| Proportion of RM tasks noted as complete | Scaled covariate | The proportion of RM tasks that are noted as complete in recovery plans | U.S. |
| First fiscal year of RM | Scaled covariate | The fiscal year the first RM task was proposed to be implemented | U.S. |
| Number of species in a recovery plan | Scaled covariate | Number of species in a multispecies recovery plan | U.S. |
| Recovery potential | Scaled covariate | Based on Recovery Priority Number 0.01 (high probability of recovery, low degree of threat), 0.16, 0.33, 0.49, 0.66, and 0.99 (low probability of recovery, high degree of threat) | U.S. |

suggesting that species with a greater historical need for information continue to require a disproportionate amount of information, or more likely, that research on a threatened species may promote interest in more research[31]. This was especially true for mammals, which arguably already have substantially more monitoring information than other taxa[32]. Fortunately, our analysis suggests that the proportion of the budget allocated to RM is decreasing over time, as the conservation community moves away from surveillance monitoring and towards more

targeted adaptive monitoring[12]. For example, the recovery plan for the Florida scrub jay (*Aphelocoma coerulescens*) was written in 1990 and management tasks were entirely RM. Since then, genetic research has demonstrated that Florida scrub jays are largely incapable of moving across habitat gaps[33]. These results have been incorporated into a new draft recovery plan, which allocates <1% of the proposed budget to ongoing research and monitoring, with the majority of resources allocated to the protection and acquisition of intact jay habitat[34].

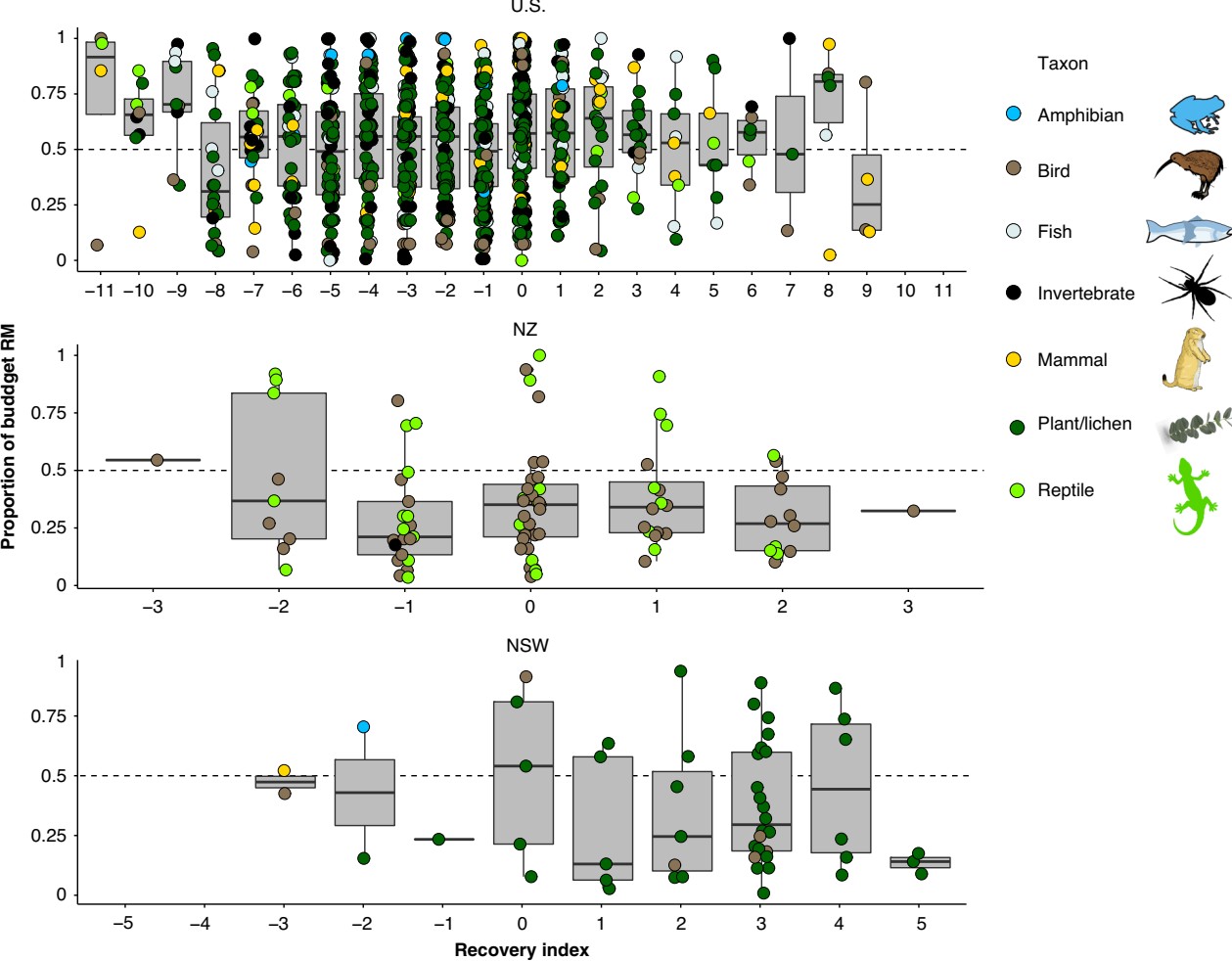

**Fig. 3 Species with poor recovery outcomes have a higher proportion of the budget allocated to research and monitoring.** The median and range of proportion of recovery plan budgets allocated to research and monitoring (RM) for $n = 1143$ threatened species in the United States (U.S.), New Zealand (NZ), and New South Wales, Australia (NSW) as a function of index of recovery. The index of recovery is the sum of reports in which population increases (+1), decreases (−1), or remains constant (0) between assessments, the range for the U.S. is: −11 to 11, NZ: −3 to 3, and NSW: −5 to 5 (although no species had an index of −5 or −4). Taxon of each species is indicated by color. For each recovery index, the box and whiskers show the median as a line, first and third quartiles as hinges, and the highest and lowest values within 1.5 times the inter-quartile range as whiskers. All vector graphics are open source.

Of all three jurisdictions, NSW had the lowest proportion of the budget allocated to RM. Here, when developing recovery plans, experts were asked only to include RM if required to inform specific management actions[17]. In recent years, NZ and NSW have assigned threatened species into streams that prioritize RM only for species where population trends, status, and threats are uncertain, and prioritize action for all species where declines are understood. Our results support the value of policies such as these that limit the allocation of resources to RM during the development of recovery plans, to establish a more effective balance between resources allocated to RM versus action.

There are numerous scientific tools that can help balance resources invested in RM and action in order to maximize the probability of achieving conservation outcomes for different species in unique contexts. This includes cost-effectiveness analysis[35] and Value of Information (VOI) analysis, which aims to improve management outcomes by understanding the optimal balance between conservation action and efficiency gained by gathering new information through RM[5,36,37]. For example, one single-species VOI study has examined the optimal allocation of resources for a threatened species (Koalas, *Phascolarctos cinereus*[6]) and found that no more than 1.7% of the recovery budget should be spent on RM. Systematic prioritization tools e.g.,[38] could also be used in sensitivity analyses to indicate areas of uncertainty that have the greatest influence on management decision-making for threatened species. Socio-economic context is also an important consideration when determining what proportion of a budget to allocate to RM to achieve conservation outcomes, where our analyses present data from relatively resource rich countries[39].

Given the ongoing biodiversity crisis, the continual shortfalls in conservation budgets, and consistent undervaluation of nature, managers are tasked with impossible decisions about how to allocate meagre conservation resources. Bending the curve for biodiversity means not just halting declines, but also recovering imperiled populations, and achieving this challenging goal will require transformative societal change[40,41]. Although much more is needed, increasing the efficiency of recovery efforts can help facilitate progress to improve outcomes for threatened species. By carefully and strategically limiting RM to that which increases our ability to deliver actions that improve the status of a species, we

can preserve resources for the implementation of actions that will ultimately recover populations.

## Methods

**Threatened species assessments.** We assessed the proportion of the proposed budget allocated to RM for a total of 2328 species, independently managed subspecies, or distinct populations (hereafter species): 700 in NZ, 361 in NSW, and 1267 freshwater and terrestrial species in the U.S. In all jurisdictions this included the most threatened listed species and/or those with recovery plans: species with Threatened and Endangered status in the U.S. with active recovery plans as of January 2017, species that met a series of criteria in NSW as of 2013 (e.g., excluding less threatened species that do not require any active intervention and those with a large geographic range[17]), and the most threatened species in New Zealand as of 2012, which included all species in the Threatened and At-Risk categories with declining populations[42]. In all three jurisdictions, species are listed for legal protection if they are at risk of extinction. Once listed, recovery planning (including proposed projects, management tasks, and budgets) documents are developed with the objective of securing species from extinction and recovering populations to a point that they can be de-listed. Although our dataset examining threatened species recovery planning is the most comprehensive to date, our data do not represent all spending on species—there are other activities for both management action and RM that occur at a sub-jurisdictional level or outside of government.

**Estimating resources allocated to RM vs action.** We gathered information on the planned costs of management tasks necessary to achieve recovery for threatened species from previously published recovery planning databases (details provided in refs. [15,16,43,44] and Supplementary Methods). Briefly, for NZ and NSW, a suite of management tasks had been developed during structured expert elicitation workshops, as part of a systematic prioritization exercise[16,17]. For the U.S., management tasks and their cost had been extracted from each species' published recovery plans (Supplementary Methods[15]). These data represent an evolution of the implementation of a systematic and cost-effective approach to endangered species resource allocation (i.e., the Project Prioritization Protocol), beginning with NZ in 2009[16], and subsequently applied to NSW in 2013[17] and the U.S. in 2016[15].

For each proposed management task we used the methods description to categorize tasks as research and monitoring or action based on the definitions in IUCN classification schemes (https://www.iucnredlist.org/resources/classification-schemes, Supplementary Table 1[45]). For NZ and NSW, using previously published datasets we used a combination of the methods description field and 4 other columns that classified the management task methods into increasingly general categories[16,17,43]. We used keywords such as *survey, monitor, surveillance, develop techniques, inventory, research*, and *develop plan* to search for research and monitoring tasks. We reviewed the management tasks identified by these broad search terms to ensure only research and monitoring tasks were included. We also reviewed the management tasks that were not captured by search terms to ensure no research and monitoring tasks were excluded. For the US, the methods descriptions were too complex for keyword searches. Instead, the first author and a trained technician classified each management task manually. To ensure that management tasks were being classified similarly, the first 200 tasks were classified by both observers and any uncertainty was flagged for review together.

For all jurisdictions, any methods descriptions that were vague, lacked context, or required further assumptions were excluded (2.6% of management tasks, U.S. only). Some management tasks (3.9%) were scored as both action and RM (e.g., translocate birds, action, and monitor the success of the release, RM; weed surveillance, RM, and control, action). For some management tasks, the distinction between action and RM was unclear. These tasks were discussed among the authors and the technician to reach a consensus. For example, 'standard surveillance to detect invasive mammals' in NZ could be considered an action, since it is required to detect and subsequently control invasions. However, we assigned it as RM because other management tasks clearly include an action component (e.g., 'surveillance for invasive species and control if detected') and other authors have categorized invasive species surveillance as monitoring[46]. Generally, management tasks to develop conservation plans are distinct from implementing plans and were thus scored as RM (K. Martin pers. comm.). Where we were unable to distinguish between RM and action, we scored as both action and RM.

For a subset of 8050 management tasks (the first 207 species) in U.S. recovery plans, we further categorized the type of RM to explore common RM activities (Supplementary Table 1). Because we found that assigning management tasks into these 17 categories was challenging without making subjective judgement calls, we did not analyze specific tasks further.

We estimated the cost of implementing each management task for each species following similar methods to those previously published, calculating costs over 50 years[15], Supplementary Methods[16,17]. We calculated the proportion of the proposed budget allocated to RM for each species as the total cost of all management tasks scored as research or monitoring divided by the total cost of all management tasks. For management tasks that were scored as both action and RM, we multiplied the cost of the task by the average proportional difference between action and RM for each jurisdiction.

**Factors affecting the proportion allocated to RM.** We compared the characteristics of each species recovery plan with the proportion of proposed spending designated as RM. Characteristics available in recovery planning databases for all three jurisdictions included taxon, the estimated benefit of implementing all management tasks, and the total budget estimated for each species (Table 1, Supplementary Methods). The most general category shared among all jurisdictions was taxon, resulting in nine categories: amphibians, birds, bryophytes, fishes, fungus, invertebrates, mammals, reptiles, and vascular plants (set as a reference category). Lichens were removed from further analysis because there were only two species. For NZ and NSW, we extracted expert-elicited estimates of the benefit of implementing all management tasks, where experts were asked to consider the probability of species being secure in 50 years with and without the suite of management tasks[16,17]. Thus, benefit was calculated as the difference between the probability of security with and without the management tasks. For the U.S., in the absence of expert elicitation, the benefit of completing all management tasks in a recovery plan was approximated using information embedded in Recovery Priority Numbers (RPN). RPNs are an 18-category numeric rank for each species based on three categories of threat (high, moderate, and low), high or low recovery potential, and taxonomic distinctness monotypic genus, species, and subspecies,[47]. The limitations of using RPN to estimate the probability of persistence with or without management are discussed by Gerber et al.[15] and Avery-Gomm[48]. To generate the total budget for each species, we used previously published total costs, which considered actions that benefited more than one species cost as shared among species projects[15–17]. In all further analysis, we removed species with a proposed budget of 0 (23 species in the U.S.) and extinct species (Guam broadbill—*Myiagra freycineti* and Eastern puma—*Puma concolor couguar*).

We explored additional characteristics unique to U.S. recovery planning documents, using U.S. data only (Table 1). These included: the federal listing status, the number of species in the recovery plan (66% of plans include multiple species), the priority assigned to each management task (1: emergency measures needed to prevent extinction, 2: measures required to stabilize a species headed for extinction, and 3: needed to delist), the estimated management task duration in years, the fiscal year the management task was implemented, the management task status (ongoing, complete, planned, discontinued), the total estimated time to recovery, and an RPN, which we used to make a new factor called 'recovery potential' (one of six scores based on the RPN, where the highest had a high probability of recovery and low degree of threat and the lowest had a low probability of recovery and a high degree of threat). Federal listing status was collapsed from six into three categories: endangered, threatened, and not listed (including candidate species, species removed from ESA due to recovery, or populations considered as 'non-essential, experimental'). Taxa were assigned to eight categories: amphibians, birds, fishes, invertebrates (set as a reference category), mammals, reptiles, and flowering and non-flowering plants.

**Quantitative analysis.** To examine what characteristics of recovery plans are associated with the proportion of the budget allocated to RM we used beta regression in the *betareg* package[49] in R version 3.6.1[50]. We fit two models—one including all data, with jurisdiction included as a covariate, and one including a wider suite of covariates only available for the U.S. (Table 1). All continuous covariates were standardized by subtracting the mean and dividing by the standard deviation to ensure the resulting parameter estimates would be comparable[51]. We standardized the total budget of each jurisdiction separately to account for each countries' different currency and year the budget was estimated. To improve model fit we removed five species with total budgets over 5 million dollars (five times the median): Barton Springs salamander - *Eurycea sosorum*, Austin blind salamander—*Eurycea waterlooensis*, Indiana bat—*Myotis sodalis*, Bull trout - *Salvelinus confluentus*, Grizzly Bear—*Ursus arctos horribilis*). Our results are robust to the inclusion or exclusion of these species.

Categorical covariates were converted to dummy variables. To select a reference category, we ran an initial model, using the category with the lowest mean proportion of budget RM as the reference. In this initial model, we selected the dummy variable with the highest variance inflation factor VIF in the car package[52]; as the reference in the final model. As a result, all VIF were <2 in final models, indicating little correlation between covariates. We found that the number of species in a recovery plan and the first fiscal year of RM were correlated with the total budget (VIF >3). We excluded correlated covariates in successive models and chose the final model with the lowest Akaike's Information Criterion (AIC[53]). The final model excluded the total proposed budget, which was correlated with the number of species in multi-species plans and the first fiscal year of the earliest RM. We consider any covariates where 95% confidence intervals around parameter estimates exclude zero to indicate a significant effect.

**Estimating species recovery outcomes.** To assess the relationship between the proportion of the budget allocated to RM and species recovery outcomes, we extracted a previously published index of recovery for U.S. listed species[2] and developed similar indices based on annual and semi-annual reports from NZ and NSW (Supplementary Methods).

To generate the U.S. recovery index, Gerber[2] calculated sums of biennial status data from reports to Congress during 1989–2011 (total of 11 status reports[30]). For each species, reports included whether their status was extinct, declining (scored as

−1), stable (scored as 0), improving (scored as +1), or unknown. These scores were summed, generating values from −11 to 11, indicating whether species are declining or improving more frequently.

To develop recovery indices for NZ and NSW, we used similar reports through the New Zealand Threat Classification System and New South Wales Saving our Species annual report card over 4 and 5 assessment periods respectively. Assessments were annual in NSW and in NZ the periods between reports were on average every 4 years (Supplementary Methods). For each update or report card, we used a similar scoring (−1, 0, and +1) to indicate whether species were declining, stable, or improving between assessments (further details in Supplementary Methods). Note that in this analysis we were limited to a subset of the 2328 threatened species (78.5% of U.S. species, 13.5% of NZ species, and 14.7% of NSW species). Other studies have noted the limitations of recovery assessments[28].

**Reporting summary**. Further information on research design is available in the Nature Research Reporting Summary linked to this article.

## Data availability

The datasets generated during and/or analyzed during the current study are available in the Figshare repository, https://doi.org/10.6084/m9.figshare.12071358.v1. Note that some unique U.S. species identifiers have been removed in compliance with USFWS. Data for recovery indices in New Zealand were extracted from the NZ Threat Classification System online database (https://nztcs.org.nz/home), for New South Wales from Saving our Species (https://www.environment.nsw.gov.au/topics/animals-and-plants/threatened-species/saving-our-species-report-cards), and for the United States from https://www.pnas.org/content/pnas/suppl/2016/03/08/1525085113.DCSupplemental/pnas.1525085113.sapp.pdf.

## Code availability

All code used for analysis during the current study are available in the Figshare repository, https://doi.org/10.6084/m9.figshare.12071358.v1.

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

## Acknowledgements

We thank G. Iacona and L. Gerber for advice on analyzing the U.S. recovery planning database, S. Davis for coding U.S. actions into categories, and R. Kaler, R. Maloney, A. Hawcroft, J. Rolfe, and K. Martin for insight into the U.S. and New Zealand recovery planning processes. J. Bennett was funded by the Natural Sciences and Engineering Research Council of Canada (NSERC) Discovery Grant #06147.

## Author contributions

R.T.B. and J.R.B. formulated and designed the study. R.T.B., S.A.G., J.R.B., and H.Y.L. collected and analysed data. R.T.B., J.R.B., S.A.G., P.A.S., S.J.C., and H.Y.L. contributed to interpreting the results and writing and editing the paper.

## Competing interests

The authors declare no competing interests.
