## [Peer Review File · Nature Communications]

REVIEWER COMMENTS

Reviewer #1 (Remarks to the Author):

This is an excellent paper. It exposes an important problem, is well written and uncovers important information for conservation practice and policy globally. The work sends an important message. It will have great impact. Simple. Clear. On point.

I have only two thoughts -

1 the papers say (a couple of times)
) - "we found that threatened species with poorer recovery outcomes had higher proportions of their recovery budgets allocated to RM. "

Notably, inferring this means more \$ spent on recovery means less success is a bit dangerous here, as there is a chicken and egg problem. This could mean that we allocate more money to recovery when we think things are going badly and we don't know what to do. Hence there may be a predisposition to spend more resources on species that are likely to do worse. This needs to be noted or acknowledged. That said, the investment in research is still excessive.

2 some research includes action - research and action can be hard to separate. this needs acknowledgement. Indeed, in reality the fraction spent on research and the fraction spent on actions to recover a species could sum to over 100%. maybe this is covered in supp material

Small comment

3 an excellent and specific paper about monitoring to extinction is by Tara Martin in Cons Letters on orange-bellied parrots and Xmas island pippistrelle

Hugh Possingham

Reviewer #2 (Remarks to the Author):

Review of NCOMMS-20-14453-T by Josie Carwardine

The manuscript assesses the proportion of species recovery budgets allocated to research and monitoring, RM, in US, NZ and NSW, Australia. This is a really informative manuscript on an important topic providing new information, the methods are very sound and I enjoyed reading it overall. However, there are a number of shortcomings that need to be addressed before the manuscript is suitable for publication. The ms doesn't clearly define RM, and it doesn't describe the work in a way that encompasses all of the types of RM, focusing instead on information collection, rather than also on improved methods that are often sought by RM. I also feel that the authors could make better use of the data, and there are a number of smaller changes required. Below are my suggestions for improvement:

1) The manuscript needs a clear definition of what RM entails and messaging that aligns to it.

The closest thing to a definition I could find is very limited - line 36: "Research and monitoring (RM) is an important component of threatened species conservation, providing information about species ecology, trends, and population biology". This doesn't allude to the development of new methods for management, nor conservation planning. However, based on the search terms for RM from SI "survey", "monitor", "surveillance", "develop techniques", "inventory", "research", "develop plan" - there is more here than just information provision.

The authors insinuate throughout the opening of the introduction and throughout much of the manuscript, that RM is about 'gaining information', which totally overlooks the research that is designed to improve technology, developing methods for threat management, etc...elements that are clearly included in RM based on the search terms. There are multiple instances talking about the need to justify the collection of information, again this is not all that RM consists of. There is another important point here which is broader than RM being about the value of information, it is also about the value of efficiency/feasibility gained by developing new approaches for management.

On a more minor note, the inclusion of 'develop plan' in the search terms indicates that the authors determined planning to be part of research and monitoring, which is broadly consistent with IUCN definitions of conservation planning being included there. Planning does create information on priorities or when to implement actions and has a value of information because it can lead to more efficiency, but I think many readers may not realise to what extent the authors included planning. A conservation planning research project that is needed to develop a new approach entirely makes sense as research, but are there some types of planning that are not part of research? What about fire management operational planning? Is developing the recovery plan itself, research? I realise that the authors went through and checked each action after doing the search, but it is still unclear as to what kinds of actions were included and which weren't.

2) The analysis could make much more of the different types of RM

Wouldn't it be interesting to understand the resource allocations between Research and Monitoring? Looking at the search terms it could be quite straightforward to split and examine this? It would also be interesting to know how many of the RM actions were linked to an objective in some way, i.e. are they targeted to achieve an outcome through improving knowledge, efficiency or technical ability to reduce a threat? Perhaps this information is not available, but this could at a minimum be acknowledged as useful information.

3) The authors need to amend unfounded assertions and assumed causalities

The manuscript contains what I think sound like assumptions and assertions that are not totally backed up by evidence. For example:

Line 45-46 "However, because RM is by definition not an intervention, its continued prioritization over management action means that programs are unlikely to meet their recovery objectives."

It is a value judgement at this stage of the ms to assert that RM is continually prioritized over management actions. Prioritized over would mean that RM is carried out or listed more than management. Do we know this is the case from this or other data? We don't know what the ideal proportion of RM to Action is, although I do agree that 50-50 seems not right! However, given it is currently 50-50, then they are currently prioritised equally in terms of financial input, not sure about numbers of actions or overall effort. Please reword this - perhaps about non-strategic/non-justified RM and inadequate funds overall being the issue?

Line 74-76: "The proportion of the budget allocated to RM was lower for species where the predicted benefits of the actions contained within recovery plans were estimated to be higher (see Methods for details, Table S2)." And lines 112 onward "Across all jurisdictions, we found that threatened species with poorer recovery outcomes had higher proportions of their recovery budgets allocated to RM." It sounds like it is being implied in the ms that lower proportional budgets allocated to RM result in higher recovery estimates, but couldn't it be the case that species with lower recovery estimates require higher proportions of RM? I understand that RM is uncertain as to whether it will provide a benefit, which would be why carrying it out may not increase recovery estimates significantly. The real question is whether, if you re-allocated those funds to undertaking action, that would increase recovery estimates for those species, either through providing new information or new more feasible techniques.

Line 122-125: "Among threatened species in the U.S., we found that when RM began longer ago there was a higher proportion of the budget allocated to RM, suggesting that research on a threatened species may promote interest in more research (Martín-López et al. 2009)."

Or this could be suggesting that species that historically had a greater need for information, continue to disproportionately need information, compared to better known species?

Line 147-139 "By carefully considering whether RM improves the ability to deliver actions that improve the status of a species, we can increase our efficiency and bend the curve for biodiversity – not just halting declines, but recovering imperiled populations."

This is one of those classic assertions made in conservation papers. Careful considering of whether RM improves ability to deliver actions will not itself 'bend the curve' and recover imperiled populations per se, but it is fair to say that this is required as part of the solution. It is more accurate to say that the implementation of adequately funded on-ground action and strategically planned RM is essential to achieving this outcome.

4) Minor suggestions

Line 48. "As a result, many local populations and species have been monitored until extinction (Lindenmayer et al. 2013)."

You could also use the pipistrelle example in Australia, to help justify the word many in here.

Line 58 "Previous work has examined the cost of threatened species recovery plans as a resource allocation problem – optimizing the trade-off between the expected benefits and costs of management tasks (Joseph et al. 2008; Brazill-Boast et al. 2018; Gerber et al. 2018)."

This could imply that these approaches traded off the benefits and costs of specific tasks/actions within recovery plans/projects, when they only compared between species recovery projects, right? Also is Joseph et al. 2008 the right reference here, or should it be 2009?

The authors could improve clarity around the terms used around the data – I understand that the NSW data are recovery projects for species within the state, not a full national recovery plan (although in many ways superior to national recovery plans because of having a consistent objective). Could the authors use the term recovery projects throughout, for all regions looked at? This avoids confusion around the term plans, with planning as part of RM. E.g. line 79 "a larger total planning budget", sounds like it might be referring to planning as part of the RM budget, rather than the project budget? The term "project budget" would be better.

Lines 106-108: "This percentage is significantly higher than research and development costs in other sectors (e.g., 8.32-24.8% of annual revenue: pharmaceutical industry, IHSP 2016; ~13% of annual revenue: top 10 largest corporations, Strategy& 2018)."

Curious as to why medical industry isn't included here, as I assume this is slightly different to pharmaceutical industry? What is IHSP, write in full. It doesn't surprise me that industries aiming to make money are not spending as much on research. I just want to make sure this doesn't come across like cherry picking.

Extra word "that" at line 122.

Line 138-140: "Scientific tools, such as Value of Information (VOI) analysis, are available to better understand the optimal balance between conservation action and new information (Canessa et al. 2015; Bennett et al. 2018)."

Another good recent reference for this is: Nicol, S., Brazill-Boast, J., Gorrod, E. et al. Quantifying the impact of uncertainty on threat management for biodiversity. *Nat Commun* 10, 3570 (2019).

<https://doi.org/10.1038/s41467-019-11404-5>

While this is true, the ms also needs to acknowledge that cost-effectiveness analysis is a useful approach to better understand the improved efficiency/feasibility/outcomes of on-ground management that could be generated by research that creates new methods.

In Figure 1 – why not shade the entire US and NZ continents the same shade of grey as the bar chart, as per NSW? I would suggest a slightly lighter grey for NSW so that the line can be seen in middle of the box and whisker plot.

Same for Figure 2.

Figure 3 – it is difficult to make out the difference in colour between some of those points for different taxa.

Reviewer #3 (Remarks to the Author):

GENERAL COMMENTS

This paper presents a detailed analysis of the relative resources spent on research and monitoring vs taking action in endangered species recovery efforts. The authors claim that a high (albeit decreasing) proportion of resources is spent on research and monitoring and that in a world of limited resources for conservation, it would be better to spend less on research and monitoring and more on taking action.

I think this is a novel, interesting and well-researched paper that should be published. As noted in my specific comments below, I have a few methodological concerns about this paper, specifically with regard to potential bias in the selection of the species studied and also the percentage of species for which the authors were able to establish a dependent variable, especially in NSW and NZ. The authors need to at least speak to these concern even if they can't 'fix' them. The authors might also want to more explicitly explore alternative hypotheses to account for the relationships that they are presenting (see my comments on Lines 12-14 below). And the authors might put in a caveat that since their data come from analyses of work in relatively resource rich countries (USA, Australia, and New Zealand) there may be challenges in extrapolating these results to other regions of the world.

Finally, I would ask the authors to consider subtly modifying their overall recommendation. As stated in more detail below, I would propose that the aim for any conservation effort should be to spend "the least amount of resources on RM that you need in order to have a reasonable chance of achieving your desired outcomes." In some cases you need to spend more on RM, in others you need to spend less....the art of doing this well is to know the right level.

SPECIFIC COMMENTS

Lines 12-14 It's important to discuss the competing hypotheses behind this statement. The authors seem to imply that more monitoring 'causes' or at least 'correlates' with negative conservation outcomes. But this could also be an effect stemming from that fact that species on the brink might legally require more extensive monitoring. Or I could think of several other hypotheses here....

Line 42 This is admittedly something of a semantic quibble – but RM is defined as a potential conservation intervention in the Conservation Measures Partnerships' taxonomy of all conservation actions, v 2.0.

(<https://docs.google.com/spreadsheets/d/1i25GTaEA80HwMvsTiYkdOoXRPWiVPZ5I6KioWx9g2zM/edit#gid=874211847>) The more important point is that RM is part of an intervention strategy – but obviously per the point of this paper – the challenge is to find the right level of RM for a given situation.

Line 65 (and ongoing) Ok, this is another quibble, but I find it a bit jarring to be reporting all means and standard deviations in this paper to the tenths of percents (one decimal place significant digits). This seems to be sending a message of false precision given the underlying data – I would be much more comfortable if no decimals were reported.

Line 97 The authors state "For species where an index of recovery could be extracted (78.5% of U.S. species, 13.5% of NZ species, and 14.7% of NSW species), those with the highest proportion of the budget allocated to RM had the lowest recovery success (Fig. 3)." Given the low % of cases with this measured dependent variable in NZ and NSW, have the authors done any analysis to ensure that this sample isn't biased? This could strongly affect major conclusions of this paper.

Line 142 I'm VERY wary of any 'rule of thumb' that tries to specify the specific percentage of resources that 'should' be spent on RM. To my mind, you need to spend the appropriate amount for the situation that you face. In some cases, you might need to spend a large percentage. In others, you might need to spend a lot less. To me the best rule of thumb (which is consistent with the message of this paper) is that you should spend "the least amount of resources on RM that you need in order to have a reasonable chance of achieving your desired outcomes." In other words, it's ALWAYS context dependent.

Line 151 The authors need to state how the sample of 2328 species was selected. There is huge potential for biased conclusions depending on how this sample was compiled. It is essential that the authors address this.

Signed - Nick Salafsky

REVIEWER COMMENTS

Reviewer #1 (Remarks to the Author):

This is an excellent paper. It exposes an important problem, is well written and uncovers important information for conservation practice and policy globally. The work sends an important message. It will have great impact. Simple. Clear. On point.

Thank you!

I have only two thoughts -

1 the papers say (a couple of times) - "we found that threatened species with poorer recovery outcomes had higher proportions of their recovery budgets allocated to RM. "

Notably, inferring this means more \$ spent on recovery means less success is a bit dangerous here, as there is a chicken and egg problem. This could mean that we allocate more money to recovery when we think things are going badly and we don't know what to do. Hence there may be a predisposition to spend more resources on species that are likely to do worse. This needs to be noted or acknowledged. That said, the investment in research is still excessive.

We agree and have added sentences to the discussion to acknowledge this dilemma (line 127):

“Second, greater allocation of resources to RM for species with poor recovery outcomes could suggest that high uncertainty associated with actions for especially imperiled species reinforces a fear of negative outcomes and may deter necessary actions (Meek et al. 2015). Thus, there may be a predisposition to spend more on RM instead of action on species that are more critically endangered. Alternatively, species with worse recovery outcomes may require higher proportions of RM, because little may be known about them and their threats. Regardless, the question remains: would reallocating funds to more action improve recovery outcomes and if so, what is the optimal allocation between RM and action to maximize the achievement of conservation objectives?”

2 some research includes action - research and action can be hard to separate. this needs acknowledgement. Indeed, in reality the fraction spent on research and the fraction spent on actions to recover a species could sum to over 100%. maybe this is covered in supp material

In some cases (3.9%) management tasks were both research/monitoring and action and were scored as such. We moved this up to line 222 along with expanded text about scoring RM and action to clarify:

“A small percentage of management tasks (3.9%) were scored as both action and RM (e.g., translocate birds (action) and monitor the success of the release (RM)).”

For any tricky cases, the authors and technician discussed and came to a consensus and any cases where the distinction was unclear were scored as both action and RM. For the particular case of developing conservation plans (see Dr. Carwardine's comment below), we checked with a USFWS manager who confirmed that developing and implementing plans are distinct management tasks in the US. This is now included in the text on line 223:

“Generally, management tasks to develop conservation plans are distinct from implementing plans and were thus scored as RM (K. Martin pers. comm.). We discussed any management tasks where RM and action were difficult to distinguish to come to a consensus and if the distinction was unclear, we scored as both action and RM (Appendix S1)”

Small comment

3 an excellent and specific paper about monitoring to extinction is by Tara Martin in Cons Letters on orange-bellied parrots and Xmas island pippistrelle

Great reference, thanks, we have inserted it on line 53.

Hugh Possingham

Reviewer #2 (Remarks to the Author):

Review of NCOMMS-20-14453-T by Josie Carwardine

The manuscript assesses the proportion of species recovery budgets allocated to research and monitoring, RM, in US, NZ and NSW, Australia. This is a really informative manuscript on an important topic providing new information, the methods are very sound and I enjoyed reading it overall. However, there are a number of shortcomings that need to be addressed before the manuscript is suitable for publication. The ms doesn't clearly define RM, and it doesn't describe the work in a way that encompasses all of the types of RM, focusing instead on information collection, rather than also on improved methods that are often sought by RM. I also feel that the authors could make better use of the data, and there are a number of smaller changes required. Below are my suggestions for improvement:

Thank you, and thanks for the thorough and thoughtful suggestions!

1) The manuscript needs a clear definition of what RM entails and messaging that aligns to it.

The closest thing to a definition I could find is very limited - line 36: “Research and monitoring (RM) is an important component of threatened species conservation, providing information about species ecology, trends, and population biology”. This doesn't allude to the development of new methods for management, nor conservation planning. However, based on the search terms for RM from SI “survey”, “monitor”, “surveillance”, “develop techniques”, “inventory”, “research”, “develop plan” – there is more here than just information provision.

The authors insinuate throughout the opening of the introduction and throughout much of the manuscript, that RM is about ‘gaining information’, which totally overlooks the research that is designed to improve technology, developing methods for threat management, etc...elements that are clearly included in RM based on the search terms. There are multiple instances talking about the need to justify the collection of information, again this is not all that RM consists of. There is another important point here which is broader than RM being about the value of information, it is also about the value of efficiency/feasibility gained by developing new approaches for management.

We agree that our definition of RM was unclear and left out research that contributes to developing management approaches. We used the IUCN guidelines (<https://www.iucnredlist.org/resources/research-needed-classification-scheme>) and the definition of ‘research’ from the ‘research-implementation’ gap literature (where research provides empirical information/evidence to practitioners, <https://doi.org/10.1111/conl.12315>).

We have altered line 37 to make our definition of RM clearer:

“Research and monitoring (RM) are important components of threatened species conservation. We define RM as activities that generate information about species (e.g., ecology, trends, population biology), threats they face, the socioeconomic context in which they occur (e.g., competing land uses), and their response to interventions, including information designed to improve management approaches”

We agree that efficiency/feasibility may be gained by developing a new approach for management and that RM may be used to improve technology or develop methods for threat management. However, these increases in efficiency won’t materialize unless the new approaches (developed through information generated by RM) are implemented, i.e., ‘action’. Hence our classification of ‘developing techniques’ as RM versus ‘develop and apply techniques’ as RM and action. To highlight this distinction we added a sentence to the introduction (line 44):

“In this way, when applied through management action, research and monitoring can lead to improved efficiency and feasibility of management approaches.”

We also altered ‘Value of Information’ sentence in the discussion to emphasize the importance of efficiency gains by gathering new information through RM (line 172):

“This includes cost-effectiveness analysis (Carwardine et al. 2019) and Value of Information (VOI) analysis, which aims to improve management outcomes by understanding the optimal balance between conservation action and efficiency gained by gathering new information through RM (Canessa et al. 2015; Bennett et al. 2018; Nicol et al. 2019).”

On a more minor note, the inclusion of ‘develop plan’ in the search terms indicates that the authors determined planning to be part of research and monitoring, which is broadly consistent with IUCN definitions of conservation planning being included there. Planning does create information on priorities or when to implement actions and has a value of

information because it can lead to more efficiency, but I think many readers may not realise to what extent the authors included planning. A conservation planning research project that is needed to develop a new approach entirely makes sense as research, but are there some types of planning that are not part of research? What about fire management operational planning? Is developing the recovery plan itself, research? I realise that the authors went through and checked each action after doing the search, but it is still unclear as to what kinds of actions were included and which weren't.

The distinction between research/monitoring and action can be unclear in some cases, which was also acknowledged by Dr. Possingham, above. We have now included text on line 223 to clarify:

“Generally, management tasks to develop conservation plans are distinct from implementing plans and were thus scored as RM (K. Martin pers. comm.). We discussed any management tasks where RM and action were difficult to distinguish to come to a consensus and if the distinction was unclear, we scored as both action and RM (Appendix S1).”

Note that most management tasks that contained “develop plan” were in the US, where developing and implementing conservation plans are distinct management tasks. However there were 25 management tasks in NSW with “develop plan”. All were scored as both action and RM because the distinction was unclear (e.g., “develop appropriate fire / forestry prescription for the species based on monitoring data that is ongoing...”), except for one (*Persoonia acerosa*) which was clearly RM (“monitor to determine if there is a response to the disease - npws have developed a management plan for phytophthora”)

2) The analysis could make much more of the different types of RM

Wouldn't it be interesting to understand the resource allocations between Research and Monitoring? Looking at the search terms it could be quite straightforward to split and examine this? It would also be interesting to know how many of the RM actions were linked to an objective in some way, i.e. are they targeted to achieve an outcome through improving knowledge, efficiency or technical ability to reduce a threat? Perhaps this information is not available, but this could at a minimum be acknowledged as useful information.

We agree that this would be useful information. However, we did not have access to a database of objectives for each species or plan among jurisdictions. We attempted to split management tasks into research and monitoring when reviewing the first 8050 tasks in US plans. We found that management tasks were too ambiguous to classify as ‘research’ and ‘monitoring,’ requiring subjective judgement calls. The IUCN criteria (<https://www.iucnredlist.org/resources/research-needed-classification-scheme>) distinguishes monitoring as something that ‘occurs over the longer term,’ which was impossible to determine for most management tasks. For example, in the US the Alabama Cave Shrimp Recovery plan, the management task: “Study and monitor hydrological patterns and groundwater withdrawal” or the NSW management task for *Coprosma inopinata* “census the plant around the waterfalls plus survey suitable habitat in the area” or the NZ management task for *Seligeria diminuta* “photo plots with 10 cm × 10 cm quadrats”.

To clarify, on line 230 in the methods we added:

“Because we found that assigning management tasks into these 17 categories was challenging without making subjective judgement calls, we did not analyze specific tasks further.”

3) The authors need to amend unfounded assertions and assumed causalities

The manuscript contains what I think sound like assumptions and assertions that are not totally backed up by evidence. For example:

Line 45-46 “However, because RM is by definition not an intervention, its continued prioritization over management action means that programs are unlikely to meet their recovery objectives.”

It is a value judgement at this stage of the ms to assert that RM is continually prioritized over management actions. Prioritized over would mean that RM is carried out or listed more than management. Do we know this is the case from this or other data? We don't know what the ideal proportion of RM to Action is, although I do agree that 50-50 seems not right! However, given it is currently 50-50, then they are currently prioritised equally in terms of financial input, not sure about numbers of actions or overall effort. Please reword this – perhaps about non-strategic/non-justified RM and inadequate funds overall being the issue?

This sentence was intended to capture the issue of non-strategic RM and inadequate funds, but it's also about spending large parts of a limited budget on RM, leaving few resources for action. We reworded this sentence to address these concerns and capture these issues (line 45):

“However, non-strategic or unwarranted RM can waste limited conservation resources, and reduces the funding available for action (McDonald-Madden et al. 2010)”

Line 74-76: “The proportion of the budget allocated to RM was lower for species where the predicted benefits of the actions contained within recovery plans were estimated to be higher (see Methods for details, Table S2).” And lines 112 onward “Across all jurisdictions, we found that threatened species with poorer recovery outcomes had higher proportions of their recovery budgets allocated to RM.”

It sounds like it is being implied in the ms that lower proportional budgets allocated to RM result in higher recovery estimates, but couldn't it be the case that species with lower recovery estimates require higher proportions of RM? I understand that RM is uncertain as to whether it will provide a benefit, which would be why carrying it out may not increase recovery estimates significantly. The real question is whether, if you re-allocated those funds to undertaking action, that would increase recovery estimates for those species, either through providing new information or new more feasible techniques.

This is a good point. We have expanded this section of the discussion to address these concerns and those of the other reviewers (line 124):

“Across all jurisdictions, we found that threatened species with poorer recovery outcomes had higher proportions of their recovery budgets allocated to RM. This relationship is likely a result of several factors. First, it suggests that planning almost exclusively for RM with little plan for action in recovery strategies is unlikely to abate threats and improve species status. Second,

greater allocation of resources to RM for species with poor recovery outcomes could suggest that high uncertainty associated with actions for especially imperiled species reinforces a fear of negative outcomes and may deter necessary action (Meek et al. 2015). Thus, there may be a predisposition to spend more on RM instead of action on species that are more critically endangered. Alternatively, species with lower recovery estimates may require higher proportions of RM, either because less is known about them or for legal reasons. Regardless, the questions remains, would reallocating funds to more action improve recovery outcomes and if so, what is the optimal allocation between RM and action to maximize the achievement of conservation objectives?”

Line 122-125: “Among threatened species in the U.S., we found that when RM began longer ago there was a higher proportion of the budget allocated to RM, suggesting that research on a threatened species may promote interest in more research (Martín-López et al. 2009).”

Or this could be suggesting that species that historically had a greater need for information, continue to disproportionately need information, compared to better known species?

This is a good point. We made the following revision (line 148):

“..perhaps suggesting that species with a greater historical need for information continue to require a disproportionate amount of information, or more likely, that research on a threatened species may promote interest in more research”

Line 147-139 “By carefully considering whether RM improves the ability to deliver actions that improve the status of a species, we can increase our efficiency and bend the curve for biodiversity – not just halting declines, but recovering imperiled populations.”

This is one of those classic assertions made in conservation papers. Careful considering of whether RM improves ability to deliver actions will not itself ‘bend the curve’ and recover imperiled populations per se, but it is fair to say that this is required as part of the solution. It is more accurate to say that the implementation of adequately funded on-ground action and strategically planned RM is essential to achieving this outcome.

To address this suggestion, we altered this sentence (line 190):

“Bending the curve for biodiversity means not just halting declines, but also recovering imperiled populations (Mace et al. 2018). Increasing the efficiency endangered species conservation approaches can facilitate progress towards achieving this challenging goal. By carefully and strategically limiting RM to that which increases our ability to deliver actions that improve the status of a species, we can preserve resources for the implementation of actions that will ultimately recover populations.”

4) Minor suggestions

Line 48. “As a result, many local populations and species have been monitored until extinction (Lindenmayer et al. 2013).”

You could also use the pipistrelle example in Australia, to help justify the word many in

here.

We now include Martin et al. 2018 outlining this example.

Line 58 “Previous work has examined the cost of threatened species recovery plans as a resource allocation problem – optimizing the trade-off between the expected benefits and costs of management tasks (Joseph et al. 2008; Brazill-Boast et al. 2018; Gerber et al. 2018).”

This could imply that these approaches traded off the benefits and costs of specific tasks/actions within recovery plans/projects, when they only compared between species recovery projects, right? Also is Joseph et al. 2008 the right reference here, or should it be 2009?

We clarified by changing to (line 55):

“costs of management (*Joseph et al. 2009,*”

The authors could improve clarity around the terms used around the data – I understand that the NSW data are recovery projects for species within the state, not a full national recovery plan (although in many ways superior to national recovery plans because of having a consistent objective). Could the authors use the term recovery projects throughout, for all regions looked at? This avoids confusion around the term plans, with planning as part of RM. E.g. line 79 “a larger total planning budget”, sounds like it might be referring to planning as part of the RM budget, rather than the project budget? The term “project budget” would be better.

We replaced instances of “recovery plan/planning budget” with “proposed budget” to maintain consistency. We wanted to avoid confusion between recovery projects, which sounds like only one management task and suggest that funding is secured, versus what we analyzed: the proposed budget for management tasks in a recovery plan. We left ‘recovery planning’ in the methods, clarifying on line 200:

“Once listed, recovery planning (*including proposed projects, management tasks, and budgets*) documents are established with the objective of securing species from extinction and recovering populations to a point that they can be de-listed.”

Lines 106-108: “This percentage is significantly higher than research and development costs in other sectors (e.g., 8.32-24.8% of annual revenue: pharmaceutical industry, IHSP 2016; ~13% of annual revenue: top 10 largest corporations, Strategy& 2018).”

Curious as to why medical industry isn’t included here, as I assume this is slightly different to pharmaceutical industry? What is IHSP, write in full. It doesn’t surprise me that industries aiming to make money are not spending as much on research. I just want to make sure this doesn’t come across like cherry picking.

We chose to highlight the pharmaceutical industry rather than other industries (e.g., medical industry) because the pharmaceutical industry is known to invest the most on R&D, and yet the

average percent is still significantly lower than 50 (line 111):

“This percentage is significantly higher than research and development (R&D) costs in other sectors: the top 10 largest corporations spend ~13% of annual revenue on R&D (Strategy& 2018), and the pharmaceutical industry, *which invests the most in R&D of any industry* (Schuhmacher et al. 2016), spends on average 8-25% of its annual revenue on R&D initiatives (Institute for Health & Socio-Economic Policy - IHSP 2016).”

Extra word “that” at line 122.

Deleted.

Line 138-140: “Scientific tools, such as Value of Information (VOI) analysis, are available to better understand the optimal balance between conservation action and new information (Canessa et al. 2015; Bennett et al. 2018).”

Another good recent reference for this is: Nicol, S., Brazill-Boast, J., Gorrod, E. et al. Quantifying the impact of uncertainty on threat management for biodiversity. Nat Commun 10, 3570 (2019). <https://doi.org/10.1038/s41467-019-11404-5>

While this is true, the ms also needs to acknowledge that cost-effectiveness analysis is a useful approach to better understand the improved efficiency/feasibility/outcomes of on-ground management that could be generated by research that creates new methods.

We added the citation and altered this sentence to reflect that VOI and cost-effectiveness analyses can lead to gains in efficiency due to RM (line 170):

“There are numerous scientific tools that can help balance resources invested in RM and action in order to *maximize the probability of achieving conservation outcomes* for different species in unique contexts. This includes cost-effectiveness analysis (Carwardine et al. 2019) and Value of Information (VOI) analysis, which aims to *improve management outcomes* by understanding the optimal balance between conservation action and *efficiency gained by gathering new information through RM*”

In Figure 1 – why not shade the entire US and NZ continents the same shade of grey as the bar chart, as per NSW? I would suggest a slightly lighter grey for NSW so that the line can be seen in middle of the box and whisker plot.

Same for Figure 2.

We fixed these figures to include these recommendations.

Figure 3 – it is difficult to make out the difference in colour between some of those points for different taxa.

We altered the color and made the points larger to be able to better distinguish between taxa.

Reviewer #3 (Remarks to the Author):

GENERAL COMMENTS

This paper presents a detailed analysis of the relative resources spent on research and monitoring vs taking action in endangered species recovery efforts. The authors claim that a high (albeit decreasing) proportion of resources is spent on research and monitoring and that in a world of limited resources for conservation, it would be better to spend less on research and monitoring and more on taking action.

I think this is a novel, interesting and well-researched paper that should be published. As noted in my specific comments below, I have a few methodological concerns about this paper, specifically with regard to potential bias in the selection of the species studied and also the percentage of species for which the authors were able to establish a dependent variable, especially in NSW and NZ. The authors need to at least speak to these concern even if they can't 'fix' them. The authors might also want to more explicitly explore alternative hypotheses to account for the relationships that they are presenting (see my comments on Lines 12-14 below). And the authors might put in a caveat that since their data come from analyses of work in relatively resource rich countries (USA, Australia, and New Zealand) there may be challenges in extrapolating these results to other regions of the world.

Thank you. We address the potential bias below and explore other hypotheses in a new paragraph in the discussion (see detailed response below). Additionally, we have put a caveat about how this analysis considers relatively resource rich countries in the discussion (line 181):

“Moreover, our analyses present data from relatively resource rich countries and socio-economic context is an important consideration when determining what proportion of a budget to allocate to RM to achieve conservation outcomes (Danielsen et al. 2003).”

Finally, I would ask the authors to consider subtly modifying their overall recommendation. As stated in more detail below, I would propose that the aim for any conservation effort should be to spend “the least amount of resources on RM that you need in order to have a reasonable chance of achieving your desired outcomes.” In some cases you need to spend more on RM, in others you need to spend less....the art of doing this well is to know the right level.

Agreed. We have adjusted the concluding paragraph to better address the art of this balance between RM and action to achieve conservation outcomes. We recommend using scientific tools to help make this art into a bit more of a science (line 170):

“There are numerous scientific tools that can help balance resources invested in RM and action in order to maximize the probability of achieving conservation outcomes for different species in unique contexts .”

SPECIFIC COMMENTS

Lines 12-14 It's important to discuss the competing hypotheses behind this statement. The authors seem to imply that more monitoring 'causes' or at least 'correlates' with negative conservation outcomes. But this could also be an effect stemming from that fact that species

on the brink might legally require more extensive monitoring. Or I could think of several other hypotheses here....

We agree that this discussion is important, thus we have added a paragraph to the discussion on line 125:

“This relationship is likely a result of several factors. First, this suggests that planning almost exclusively for RM with little plan for action in recovery strategies is unlikely to abate threats and improve species status. Second, greater allocation of resources to RM for species with poor recovery outcomes could suggest that high uncertainty associated with actions for especially imperiled species reinforces a fear of negative outcomes and may deter necessary conservation actions (Meek et al. 2015). Thus, there may be a predisposition to spend more on RM instead of action on species that are more critically endangered. Alternatively, species with worse recovery outcomes may require higher proportions of RM, because little may be known about them and their threats.”

After reviewing the Biodiversity Conservation Act (NSW: <https://legislation.nsw.gov.au/#/view/act/2016/63/>), Endangered Species Act (<https://www.fws.gov/endangered/esa-library/pdf/ESAall.pdf>), and the New Zealand Threat Classification system (<https://www.doc.govt.nz/Documents/science-and-technical/sap244.pdf>) we were unable to find evidence that critically endangered species legally require more extensive monitoring, thus we excluded this hypothesis.

Line 42 This is admittedly something of a semantic quibble – but RM is defined as a potential conservation intervention in the Conservation Measures Partnerships’ taxonomy of all conservation actions, v 2.0. (<https://docs.google.com/spreadsheets/d/1i25GTaEA80HwMvsTiYkdOoXRPWiVPZ5l6KioWx9g2zM/edit#gid=874211847>) The more important point is that RM is part of an intervention strategy – but obviously per the point of this paper – the challenge is to find the right level of RM for a given situation.

Thank you for the new taxonomy. We have altered this section (line 45):

“However, non-strategic or unwarranted RM can waste limited conservation resources, and reduces the funding available for action (McDonald-Madden et al. 2010).”

Line 65 (and ongoing) Ok, this is another quibble, but I find it a bit jarring to be reporting all means and standard deviations in this paper to the tenths of percents (one decimal place significant digits). This seems to be sending a message of false precision given the underlying data – I would be much more comfortable if no decimals were reported.

Thanks for your feedback. We have removed all decimal points.

Line 97 The authors state “For species where an index of recovery could be extracted (78.5% of U.S. species, 13.5% of NZ species, and 14.7% of NSW species), those with the highest proportion of the budget allocated to RM had the lowest recovery success (Fig. 3).”

Given the low % of cases with this measured dependent variable in NZ and NSW, have the authors done any analysis to ensure that this sample isn't biased? This could strongly affect major conclusions of this paper.

The reason these percentages are so low for NZ and NSW is that we tried to capture species with several years of assessment to look at recovery indices over time. Very few species in NZ and NSW had multiple years (>3 and >5 respectively) of status assessment. Unfortunately we were limited by these assessments. Other authors have noted that, although these are the only available estimates of recovery, these indices are subjective and of relatively poor quality (e.g., Gibbs and Currie 2012 - 10.1371/journal.pone.0035730).

We now acknowledge this limitation on line 298 of the methods:

Note that in this analysis we were limited to a subset of the 2328 threatened species (78.5% of U.S. species, 13.5% of NZ species, and 14.7% of NSW species). Other studies have noted the limitations of recovery assessments (Gibbs and Currie 2012).

We examined the distribution of proportion of budget allocated to RM, which look similar for the US and NSW (2 left columns):

The distribution looks quite different for NZ (third column, upper and lower rows), skewing towards the left, meaning there are more species with a low proportion of RM compared to the entire dataset.

Line 142 I'm VERY wary of any 'rule of thumb' that tries to specify the specific percentage of resources that 'should' be spent on RM. To my mind, you need to spend the appropriate amount for the situation that you face. In some cases, you might need to spend a large percentage. In others, you might need to spend a lot less. To me the best rule of thumb (which is consistent with the message of this paper) is that you should spend "the least

amount of resources on RM that you need in order to have a reasonable chance of achieving your desired outcomes.” In other words, it’s ALWAYS context dependent.

We agree, and this is why VOI analyses are so powerful – they are applicable to different species in different contexts. We have restructured the beginning of this paragraph to clarify (line 170):

“There are numerous scientific tools that can help balance resources invested in RM and action in order to maximize the probability of achieving conservation outcomes for different species in unique contexts .”

Line 151 The authors need to state how the sample of 2328 species was selected. There is huge potential for biased conclusions depending on how this sample was compiled. It is essential that the authors address this.

We used all threatened and endangered terrestrial and freshwater species, independently managed subspecies, or distinct populations with active recovery plans in the US (from Gerber et al. 2018), a subset of listed species in NSW based on criteria outlined in Brazill-Broast et al. 2018 (excluded: those with insufficient data or expert knowledge; those that do not require any active intervention or investment, those with a large geographic range/highly mobile or dispersed, and those with less than 10% of their total population within NSW), and all species in the “threatened” and “at risk” categories with declining populations in NZ as of 2012. In both NZ and NSW, the list of species was provided by managers for planning (Brazill-Broast et al. 2018, Bennett et al. 2015 and 2017). Thus, in all jurisdictions we assessed the most threatened species with recovery plans. We added this to the methods on line 194:

“In all jurisdictions this included the most threatened listed species and/or those with recovery plans: species with Threatened and Endangered status in the U.S. with active recovery plans as of January 2017 (Appendix S1), species that met a series of criteria in NSW as of 2013 (e.g., excluding less threatened species that do not require any active intervention and those with a large geographic range; Brazill-Boast et al. 2018), and the most threatened species in New Zealand as of 2012, which included all species in the “Threatened” and “At-Risk” categories with declining populations (Bennett et al. 2014)”

REVIEWERS' COMMENTS:

Reviewer #1 (Remarks to the Author):

The authors have dealt well with my comments

Reviewer #2 (Remarks to the Author):

Overall the authors have improved the manuscript well in response to reviewers suggestions. I have a few final points which I think are important in ensuring that the angle of this paper is fair. At the moment I feel like the paper still goes in with a point to make that R and M resources are too high. However it is not up to scientists to say what is the right proportion, it is up to the science to speak for itself. The point of this paper should be that we currently don't have the science to know this proportion because we do not analyse VoI of R and M to strategically allocate funds amongst these activities. That is what the paper should firstly be calling for.

Can the authors make it clear whether the development of new technologies is included in their definition of R and M? Right now it is as though R and M is only about improved information. However R and D can still be an important part of techniques to improve management interventions?

Further to this, I still find the comparison with R and D in other sectors a bit tricky. Here the comparison is between the proportion of R and M of a conservation budget vs proportion of R and D of total revenue. Is this fair given that conservation doesn't make money (which is as a result of the environment not being valued)? Maybe it would be fairer to compare R and M and proportion of total contributions of threatened species to human society? (not that this is possible, but can it at least be acknowledged that there are other differences between these sectors?)

Finally, why doesn't the paper clearly call out the need for increased threatened species management funds? It feels a bit victim blaming to suggest the conservation community is misspending its grossly inadequate resources. And it is also very optimistic to suggest that strategically planned R and M vs action is what we ultimately need to make things better. The reality is that we do need all that funding for R and M, and we need a bunch more for implementation, and we need a huge swathe of changes to how our society conducts its operations that create all these threats to species in the first place. The tone of the concluding paragraph is a bit hard to swallow, implying that prioritisation is the main key to the solution. I think its potentially damaging to the conservation community when we are painted as failures in an incredibly under-funded system - let's be a bit kinder to each other and make sure we also call out the real problems :-). This is the second very valuable point that the paper could make.

Thanks!

Reviewer #3 (Remarks to the Author):

I am satisfied that the authors have suitably addressed the concerns raised in the first round of reviews.

This article should now be published.

Overall the authors have improved the manuscript well in response to reviewers suggestions. I have a few final points which I think are important in ensuring that the angle of this paper is fair. At the moment I feel like the paper still goes in with a point to make that R and M resources are too high. However it is not up to scientists to say what is the right proportion, it is up to the science to speak for itself. The point of this paper should be that we currently don't have the science to know this proportion because we do not analyse VoI of R and M to strategically allocate funds amongst these activities. That is what the paper should firstly be calling for.

Can the authors make it clear whether the development of new technologies is included in their definition of R and M? Right now it is as though R and M is only about improved information. However R and D can still be an important part of techniques to improve management interventions?

Thanks for this important point. In our manuscript, we define RM as generating information which can be applied to developing new technologies. In the context of our study, applying the new technologies would be classified as 'action'. The 'RM' component is generating information about the technology (e.g., how effective is it?). To clarify, we add the following to line 49:

"We define RM as activities that generate information about species (e.g., ecology, trends, population biology), threats they face, the socioeconomic context in which they occur (e.g., competing land uses), their response to interventions, and the effectiveness of new management techniques, including information designed to improve management approaches ⁷"

Further to this, I still find the comparison with R and D in other sectors a bit tricky. Here the comparison is between the proportion of R and M of a conservation budget vs proportion of R and D of total revenue. Is this fair given that conservation doesn't make money (which is as a result of the environment not being valued)? Maybe it would be fairer to compare R and M and proportion of total contributions of threatened species to human society? (not that this is possible, but can it at least be acknowledged that there are other differences between these sectors?)

We agree that this comparison, although parallel, is not equal and the undervaluing of natural capital is at the heart of the issue. As such, we added the following in Line 127:

"We note that this comparison is not direct - conservation does not typically generate revenue - and percentages would be considerably different if RM were compared to contributions of threatened species to human society, which are consistently undervalued (Guerry et al. 2015)."

Finally, why doesn't the paper clearly call out the need for increased threatened species management funds? It feels a bit victim blaming to suggest the conservation community is mispending its grossly inadequate resources. And it is also very optimistic to suggest that strategically planned R and M vs action is what we ultimately need to make things better. The reality is that we do need all that funding for R and M, and we need a bunch more for

implementation, and we need a huge swathe of changes to how our society conducts its operations that create all these threats to species in the first place. The tone of the concluding paragraph is a bit hard to swallow, implying that prioritisation is the main key to the solution. I think its potentially damaging to the conservation community when we are painted as failures in an incredibly under-funded system - let's be a bit kinder to each other and make sure we also call out the real problems :-) This is the second very valuable point that the paper could make.

Thanks!

Thanks a lot for this crucial point of view. We totally agree, and it wasn't our intention to point fingers at anyone. Instead, we were trying to emphasize that managers are tasked with incredibly challenging decisions about how to spend the meagre resources they are given to protect threatened species. The fact that large proportions of resources are allocated to RM is a by-product of the undervaluation of nature – where RM is likely to lead to less controversy than action, especially in the context of charismatic species that overlap with challenging socio-economic situations. We have revised our final paragraph to reflect our shared perspectives:

“Given the ongoing biodiversity crisis, the continual shortfalls in conservation budgets, *and consistent undervaluation of nature, managers are tasked with impossible decisions about how to allocate meagre conservation resources.* Bending the curve for biodiversity means not just halting declines, but also recovering imperiled populations, *and achieving this challenging goal will require transformative societal change*^{40,41}. *Although much more is needed,* increasing the efficiency of recovery efforts can help facilitate progress to improve outcomes for threatened species. By carefully and strategically limiting RM to that which increases our ability to deliver actions that improve the status of a species, we can preserve resources for the implementation of actions that will ultimately recover populations.”